# Research on an evolutionary game model and simulation of a cluster innovation network based on fairness preference

**Chuanyun Li**[1], **Xia Cao**[1]*, **Ming Chi**[2]

**1** Economics and Management School, Harbin Engineering University, Harbin, China, **2** Management School, Jilin University, Changchun, China

* caoxia@hrbeu.edu.cn

**Data Availability Statement:** All data needed to replicate all of the figures, graphs, tables, statistics, and other values are provided within the https://dx.doi.org/10.17504/protocols.io.8pdhvi6.

## Abstract

The cluster innovation network is an important part of regional economic development. In addition, the fairness preference of internal innovators in the processes of investment and benefit distribution are particularly important for curbing "free riding" and other speculative behaviors and for creating a good cooperation environment. Therefore, taking a cluster innovation network constructed by the weighted evolutionary BBV model as the research subject, this paper constructs an evolutionary game model of a cluster innovation network based on a spatial public goods game and the theory of fairness preferences, which involves the processes of investment and payoff allocation. Using simulation analysis, this paper studies the evolution of innovators' cooperative behaviors and benefits in cluster innovation network under the conditions of a fairness preference and a return intensity. The results show that an increase in the weight coefficient, gain coefficient and degree of differentiation between the previous income and current investment can effectively promote improvements in the level of enterprise cooperation. Indeed, the greater the weight coefficient, the gain coefficient and the degree of differentiation are, the more substantial the improvement in the level of enterprise cooperation will be. Moreover, an improvement in the differentiation of the breadth and depth of enterprise cooperation has an inhibitory effect on enterprise cooperation. Furthermore, whereas increases in regulation and gain coefficients can effectively promote enterprise cooperation. However, the increase in the weight coefficient has a different effect on enterprise benefit in terms of the breadth and depth of cooperation. Finally, we hope to improve the overall cooperation level and cooperation income of the network by deeply understanding the fair preferences of innovators in the processes of investment and benefit distribution, which is helpful for promoting the evolution and development of cluster innovation networks.

## 1 Introduction

Cluster innovation networks are an innovative organizational form designed to accelerate the development of cooperative innovation and to improve knowledge level and innovation

**Funding:** This work was financially supported by the National Natural Science Foundation of China (Grant No. 71473055).

**Competing interests:** The authors have declared that no competing interests exist.

ability, which have become indispensable part of China's regional innovation system. Since clusters possess the characteristics of geographical proximity and knowledge spillovers[1], the knowledge and technology of innovators exhibit the attributes of "public goods"[2]. In the process of cooperative innovation, speculation[3] such as "hitchhiking" and "betrayal" often occurs, and it seriously damages the cooperative environment of the cluster innovation network. Additionally, the fairness preference[4] and moderate return intensity[5] in the process of investment and the payoff allocation of innovators can effectively inhibit the emergence of these speculative activities. In addition, in the actual network evolution game process, the cooperation behavior of innovators is not only influenced by the network structure but also closely related to the intensity of the cooperative relationship between innovators[6], that is, the connection in the real network has the weight attribute. According to the existing research, a real network with connected weights has both the power-law distribution characteristics of the degree distribution and the power-law distribution characteristics of the strength distribution[7]. The weighted scale-free network constructed based on the weighted evolutionary BBV model can simulate a real network very well [8]. Therefore, we take a weighted scale-free cluster innovation network as the research subject. By considering the interaction between cooperative behavior of innovators and the network structure and analyzing the evolution of the cluster innovation network under the given fairness preferences and return intensity, we can avoid problems such as hitchhiking and cooperation inertia(among others), and improve both overall level of cooperation and the cooperative benefit of the network. This result is very significant for promoting the evolution and development of a cluster innovation network.

From the perspective of game theory, the cooperative game process among innovators in cluster innovation networks can be regarded as a spatial public goods game[9]. Because the betrayer's income is often higher than the cooperator's income, the cooperative dilemma of the "tragedy of the commons" occurs[10]. To resolve this dilemma, some scholars have found that volunteer[11], reputation[12], reward[13] and punishment[14] mechanisms can improve the level of cooperation in the network. Some scholars have also found that fixed static network structures such as rule networks[15], small-world networks[16] and scale-free networks[17] can also promote cooperation under certain conditions[18]. Santos et al [19] discovered that the fairness preference of innovative subject cooperative behavior can greatly affect the level of cooperation in BA network. Many scholars have begun to supplement and improve fairness preference theory in the context of a spatial public goods game on the network in three main ways. First, scholars seek to improve the investment process in a spatial public goods game and to study the influence of the investment fairness preference of innovators on the cooperation levels in the network. For example, some researchers have improved the investment fairness preference according to the degree value of the game subject[20–21] and have found that in the static rule network, a high-quality group preference can greatly enhance the cooperation level of the innovators. Other scholars have improved the investment fairness preference according to the previous income of the game subject[22] and have found that in the static BA network, the degree of investment differentiation increases, which can promote the cooperation level in the network. In addition, some studies have improved the investment fairness preference according to the cooperation proportion of the game subject in the neighborhood[23–25]. These studies that in the static rule network, a small increase in the investment heterogeneity can rapidly increase the cooperation level in the network. Second, studies have sought to improve the payoff allocation process in the spatial public goods game and have examined the impacts of payoff allocation fairness preferences on the cooperation levels in the network. For example, some researchers improved the payoff allocation fairness preferences according to the degree value of the game subject[26]. These researches found that in a static BA network, when the degree of

differentiation in the payoff allocation is low, the level of cooperation in the network can be higher. Other scholars have improved the payoff allocation fairness preference according to the cooperation proportion in the neighborhood of the game subject[27]. These scholars found that in a static BA network, the greater the degree of differentiation in the payoff allocation is, the higher the level of cooperation in the network will be. In addition, some scholars improved the payoff allocation fairness preference according to the previous income of the players[28] and have found that in a static BA network, the level of cooperation in the network will increase significantly only when the degree of differentiation in the distribution of interests is above a certain threshold. Third, the existing research has improved the investment and payoff allocation process in the spatial public goods game, and it studied the impacts of the investment and payoff allocation fairness preference of innovators cooperative behavior on the cooperation level and cooperative benefit in a network. For example, some researchers have improved the investment and payoff allocation according to the degree value and the current investment the game subject[29]. They have found that in static BA networks, when the innovators' neighbors allocate excessive benefits, the level and benefits of cooperation in the network can be significantly improved. Some scholars have improved the investment and payoff allocation according to the previous income and current investment[30]. These scholars found that in static regular networks, an increase in the investment differentiation can promote cooperation and increase the cooperative benefits in a network. In addition, other researchers improved the investment and payoff allocations according to the degree value, previous income, current investment and degree value[31]. These researchers found that in static BA networks, moderate enterprise degree can promote the formation of an interest community. Moreover, the improvement in the gain level is an important source of the increase in the average income and the emergence of cooperation.

In light of the existing research, the research on evolutionary games in cluster innovation networks based on the fairness preference has been substantial. Most studies are based on an established static network structure and analyze the evolution of innovators' cooperative behavior under fairness preference in the spatial public goods game. However, the existing research still has the following shortcomings: (1) it has been under an established static network structure, and does not consider dynamic changes in the network structure; (2) because scholars mostly use weightless networks (such as rule networks, small world networks and BA scale-free networks) as the network model, weighted networks with a network relationship strength are seldom examined; (3) while existing research mainly improves the rules of payoff allocation from the degree value, current investment and cooperation proportions, it ignores the impact of the intensity of cooperation among innovators on the process of payoff allocation; and (4) last, scholars mostly aim to improve the level of cooperation in the network through the improvement of the fairness preferences for the investment and payoff allocation Thus, scholars have neglected to consider the return intensity and the cooperative benefits of innovators. In light of these limitations, the present paper takes the cluster innovation network constructed by the weighted evolutionary BBV model as the research object and, based on the spatial public goods game model and fairness preference theory, constructs an evolutionary game model of the cluster innovation network that combines the process of investment and payoff allocation. Using the MATLAB 2017b software, this research simulates and analyzes the evolution of cooperative behavior and the cooperative benefits of innovators in cluster innovation networks under a fairness preference and return intensity, which has important theoretical significance and practical relevance for revealing the evolution mechanism of cluster innovation networks and promoting their development.

The remainder of this paper is organized as follows. The model with network evolution analysis is presented in Section 2. Then, the model with heterogeneity of both the investment

and payoff allocations is constructed in Section 3. Subsequently, the corresponding simulation results are given in Section 4. Finally, the conclusions are provided in Section 5.

## 2 Evolution analysis of the cluster innovation network under the fairness preference

Innovators in cluster innovation networks often show strong preferences for fairness in the process of cooperation[4], as they seek to maximize their own payoffs but also consider the fairness of the investment and payoff allocation[31]. The investment and payoff allocations are complementary processes that are independent and interrelated; that is, investment is an important prerequisite for the payoff allocation, and in turn, the payoff allocation is the main reference for the next round of investment. Therefore, from the perspective of the fairness preferences in the process of investment and payoff allocation, this paper analyzes the evolution process of cluster innovation networks.

The fairness preference of innovators for cooperative behavior is related to the scale of innovators and the intensity of their cooperative relationships. Among these factors, the strength and innovation abilities of innovators at different scales are different. The scale of the innovators greatly affects their investment fairness preferences, which can be reflected by the cooperation breadth of the nodes in cluster innovation networks. The cooperative R&D capability, trust and knowledge transfer efficiency of different cooperation intensity among innovators are also different. The cooperation intensity among innovators greatly affect the process of the payoff allocation of innovators, which can be reflected by the cooperation depth between the nodes in cluster innovation networks. Based on the relevant literature[31], to reflect the degree of differentiation in the fairness preference between the investment and payoff allocation and the importance degree of each index this paper uses the adjustment coefficient to reflect the different degree of fairness preference between the investment and payoff allocation in terms of the previous income, current investment, cooperation breadth and cooperation depth. Furthermore, the weight coefficient is used to reflect the degree of importance of the fairness preference between the investment and payoff allocation in the previous income, current investment, cooperation breadth and cooperation depth. In addition, since the return intensity is an important factor that affect the cooperative behavior and the cooperative income of the innovators in the process of the game, this paper uses the gain coefficient to adjust the return intensity from the investment cost to the income of the innovators in the process of the cooperative game [32].

In the process of the cooperative game, innovators in cluster innovation networks will adjust their cooperative strategies according to the Fermi rule [33], then change their cooperative behavior. At the same time, network nodes will adjust their cooperative goals according to the reconnection mechanism with preferred connections [34], which will change the network structure. As a result of the interaction between the network structure and innovators' cooperative behavior, the cluster innovation network's structure exhibits dynamic evolution.

Accordingly, this paper constructs an evolutionary analysis framework of a cluster innovation network under the given fairness preferences, as shown in Fig 1. By embedding the fairness preference of innovators' cooperative behavior into the game model of the process of investment and payoff allocation, and under the influence of the fairness preferences and return intensity, this paper reveals the evolution of cooperative behavior and the income of innovators in the dynamic change process of a cluster innovation network. It is of vital to promote the evolution and development of cluster innovation networks.

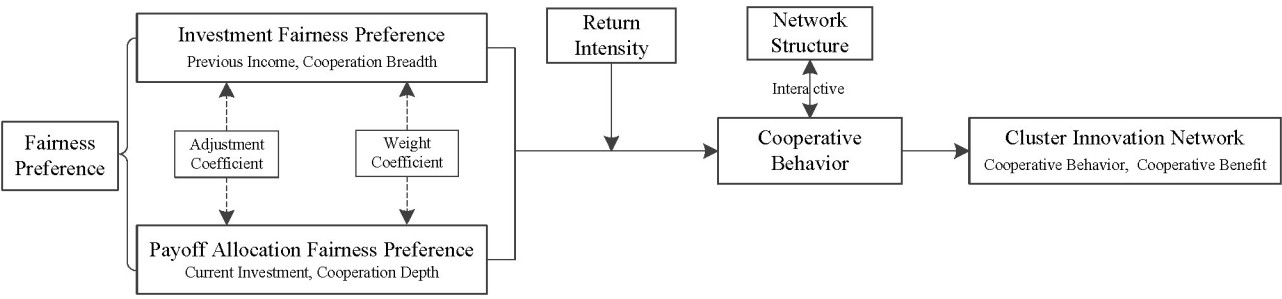

**Fig 1. Evolution analysis framework of cluster innovation networks under fairness preferences.**

## 3 Evolutionary game model of a cluster innovation network based on fairness preferences

### 3.1 Basic hypothesis of the game model

In cluster innovation networks, nodes represent cluster enterprises, links represent the game relationships between innovators, and link weights represent the strength of the cooperative relationships between innovators. The innovators in the network update their own strategies according to the rules and adjust their relationships through the reconnection mechanism with preferential connections until the cooperative strategies of the innovators and the relationships between them reach a stable state. Based on the characteristics of the cluster innovation network structure and a realistic consideration of the game model, the following hypotheses are proposed:

Hypothesis 1: In the process of the game, the cooperative relationships between innovators are adjusted only according to the reconnection mechanism with preferential connections, the adjustments are made without considering the growth of nodes and edges in the network.

Hypothesis 2: In the process of the game, the innovators $x$ with degree value $k_x$ only participate in the game between the neighborhood centered on itself and neighborhood centered. Moreover, there is a total degree value of $k_x + 1$ in the neighborhood, and all of the nodes in the network have the same total investment $C = 1$.

Hypothesis 3: The innovators in the network are all limited rational individuals and can only choose two strategies: cooperation and noncooperation.

### 3.2 Construction of the game model

In the first round ($t_n = 1$) of the public goods game, the degree value of enterprise $x$ is $k_x$, so if enterprise $x$ chooses an uncooperative strategy ($S_x = 0$), the investment of enterprise $x$ in its neighborhood is 0. However, if enterprise $x$ chooses a cooperative strategy ($S_x = 1$), it participates in $k_x + 1$ neighborhoods centered on itself and neighbors, and its total investment is evenly distributed among all $k_x + 1$ neighborhoods. At this time, the investment in the neighborhood centered on enterprise $y$ of $x$ is as follows:

$$I_{x,\hat{y}}(t_n) = \frac{1}{k_x + 1}, \qquad \text{if } S_x = 1 \text{ and } t_n = 1 \tag{1}$$

In the process of the public goods game in round $t_n$ ($t_n \geq 2$), if enterprise $x$ chooses cooperative strategy ($S_x = 1$), then it allocates the investment of one of its neighborhood $\hat{y}$ (the

neighborhood composed of neighborhood enterprise $y$ as the center and directly connected enterprises) according to Eqs (2) and (3):

$$I_{x,\hat{y}}(t_n) = C \bullet \bar{D}_{x,\hat{y}}(t_n) \tag{2}$$

$$D_{x,\hat{y}}(t_n) = (1 - w_1) \frac{m_{x,\hat{y}}^{\alpha_1}(t_{n-1})}{\sum\limits_{i=0}^{k_y} m_{i,\hat{y}}^{\alpha_1}(t_{n-1})} + w_1 \frac{k_x^{\alpha_1}}{\sum\limits_{i=0}^{k_y} k_i^{\alpha_1}}, \qquad \text{if } S_x = 1 \text{ and } t_n \geq 2 \tag{3}$$

Among these variables, because that the total investment of each enterprise is 1, $D_{x,\hat{y}}(t_n)$ is normalized according to the min-max normalization method. $\bar{D}_{x,\hat{y}}(t_n)$ is the normalized value of $D_{x,\hat{y}}(t_n)$, $t_n$ is the number of rounds of the public goods game (only if all of the nodes in the network have a specific round of the game to end the round of game), $I_{x,\hat{y}}(t_n)$ is the input of enterprise $x$ to neighborhood $\hat{y}$ in round the $t_n$ of the public goods game, and $m_{i,\hat{y}}(t_{n-1})$ is the revenue of enterprise $i$ from neighborhood $\hat{y}$ after the first round $t_{n-1}$ of the public goods game. and $m_{x,\hat{y}}(t_{n-1})$ is the revenue of enterprise $x$ from neighborhood $\hat{y}$ after the first round $t_{n-1}$ of the public goods game. When $i = 0$ denotes enterprise $y$ itself, $k_y$ denotes the degree value of enterprise $y$. Eq (3) shows that the investment of enterprise $x$ in neighborhood $\hat{y}$ is mainly measured by the previous income and degree value. Among these values, the profitability of neighborhood $\hat{y}$ is expressed by the previous income. If enterprise $x$ can obtain more profits than other enterprises in neighborhood $\hat{y}$, then enterprise $x$ will invest more in neighborhood $\hat{y}$. The degree value represents the social status of enterprise $x$ in neighborhood $\hat{y}$. Enterprises with higher degree value will earn more investment in the next round of the game. To measure the importance of the previous income and degree value in the process of enterprise investment, the weight coefficients 1-$w_1$ and $w_1$ are used, where $0 \leq w_1 \leq 1$. Moreover, to reflect the degree of differentiation of the previous income and degree value in the process of enterprise investment in the network, this paper uses the adjustment coefficient $\alpha_1$ [21]. When $\alpha_1 > 0$, this implies greater prophase income and greater proportion of the degree value, hence more investment is made in the neighborhood. When $\alpha_1 = 0$, the model is consistent with the classical public goods game, and the input is distributed equally according to the number of neighborhoods[35].

In round $t_n$ of the game, when the input of enterprise $x$ into its neighbor $y$ is over, the profits from neighbor $y$ are distributed according to Eq (4):

$$m_{x,\hat{y}}(t_n) = r[(1 - w_2) \frac{I_{x,\hat{y}}^{\alpha_2}(t_n)}{\sum\limits_{i=0}^{ky} I_{i,\hat{y}}^{\alpha_2}(t_n)} + w_2 \frac{G_x^{\alpha_2}}{\sum\limits_{i=0}^{ky} G_i^{\alpha_2}}] \bullet \sum\limits_{i=0}^{ky} I_{i,\hat{y}}(t_n) \bullet S_i(t_n) - I_{x,\hat{y}}(t_n) \bullet S_x(t_n) \tag{4}$$

Among these variables, $r$ is the gain coefficient that used to measure the return intensity of investment ($r > 1$), and $\sum\limits_{i=0}^{ky} I_{i,\hat{y}}(t_n) \bullet S_i(t_n)$ is used to represent the total investment of all enterprises $i$ in neighborhood $\hat{y}$. Eq (4) shows that the payoff allocation of enterprise $x$ in neighborhood $\hat{y}$ is mainly measured by the current investment and the strength of the cooperative relationships. Among these factors, the current investment reflects the investment ability in neighborhood. The greater the investment of enterprise $x$ in neighborhood $\hat{y}$ is, the more income enterprise $x$ will earn from neighborhood $\hat{y}$. The strength of the cooperative relationship indicates the degree of close cooperation in the relationship. The stronger the cooperative relationship between the enterprises and their neighbors is, the more profits they will earn from

their neighbors. To measure the importance of the current investment and cooperation intensity in payoff allocation, the weight coefficients $1-w_2$ and $w_2$ are used, respectively. In addition, $\alpha_2$ is the adjustment coefficient, which is consistent with the meaning and function of $\alpha_1$.

Thus, after the end of round $t_n$ of the game, the total revenue obtained by enterprise $x$ is the sum of its revenue obtained in neighborhood $k_x + 1$, that is:

$$M_x(t_n) = \sum_{\hat{y}=0}^{kx} m_{x,\hat{y}}(t_n) \tag{5}$$

### 3.3 Evolution rules

Node $m$ in the cluster innovation network will randomly select a neighbor node $n$ after each round of the game to compare strategies. If $pr_n > pr_m$, node $m$ will imitate neighbor $n$'s game strategy in the next round of the game with probability $W$. According to the Fermi update rule [33], the imitation probability is as follows:

$$W_{m \to n} = \frac{1}{1 + \exp[(pr_n - pr_m)/k]} \tag{6}$$

Here, $k$ represents the intensity of the noise, that is, the interference of external factors on the strategy learning process. When $k \to 0$, the external factors will not interfere with the node's strategy learning; on the contrary, the node can only update its strategy randomly because of the external factors. Considering the impact of the node revenue and strategy, this paper selects a neutral noise factor $K = 0.5$ as the simulation parameter value.

When node $m$ selects the strategy of learning neighbor node $n$ with probability $W$, it will be reconnected with other non neighbor nodes in the network with probability $\Upsilon_{ms}$. In the process of reconnection, only one edge is broken at a time; that is, the weight of the edge is reduced by 1. Since the nodes have certain preferences when choosing partners, this paper uses the reconnection mechanism with preferential connections [34] to determine the outgoing connection $s$ of node $m$, The random probability is as follows:

$$\Upsilon_{ms} = \sum_{m \in G} \frac{p_s^\beta}{p_m^\beta} \tag{7}$$

Here, $p_s$ is the benefit of node $s$, $G$ is the set where node $m$ is located, $\beta$ is the preference tendency, and $\beta = 0$ is the non preference connection tendency, that is, a random connection. Conversely, the preference connection tendency is greater. This paper utilizes a high preference of $\beta = 1$ for simulation.

## 4 Simulation analysis of a cluster innovation network evolution game

### 4.1 Simulation steps

Step 1: Initialize the evolutionary game parameters, and according to the cluster innovation network, randomly assign the two game strategies of "cooperation" and "noncooperation" to each node in the network, with an initial cooperation level of 50%; that is, the network cooperation density set at 0.5.

Step 2: In each round, all of the innovators play the game with their neighbors and accumulate the cooperative benefits of the innovators according to the game model.

Step 3: In each round of the game, all of the innovators update their strategies according to the Fermi strategy rule (Eq (6)) and adjust their partners based on the reconnection mechanism with preferential connections (Eq (7)).

Step 4: Repeat steps 2 and 3 until the number of Monte Carlo iterations is reached and the simulation is completed.

## 4.2 Setting and explaining of the simulation parameters

According to the evolutionary game model and the specific algorithm of cluster innovation networks, and using the simulation platform of MATLAB 2017b, we set the simulation parameters for the evolutionary game of the cluster innovation network. These parameters are given in Table 1.

In this paper, we use the generation mechanism of the weighted evolution BBV model to produce a cluster innovation network with $N = 100$ nodes and an average degree of 4. The maximum node degree is 31, the minimum node degree is 2, the maximum cooperation intensity is 10, and the minimum cooperation intensity is 1. Each data point is the average of the simulation results after 200 independent experiments. To ensure the accuracy of the research results, this paper sets the number of game rounds to 500. After the system fully evolves to a stable stage, the average of the last 50 cooperation densities is taken as Fc. Since the investment rules and payoff allocations are complementary processes that promote each other, their consistency should be maintained in the game. This paper assumes that the adjustment coefficient $\alpha_1$ of inputs and the adjustment coefficient $\alpha_2$ of the payoff allocations have the same trend of change. Moreover, the weight coefficients $w_1$ and $w_2$ also have the same trend of change, i.e., $\alpha = \alpha_1 = \alpha_2$ and $w = w_1 = w_2$. In addition, to facilitate the analysis of the effect of the weight coefficients of the fairness preferences on cooperative behavior and the cooperative benefits of the innovators in the network, this paper uses Ruguo [31] and Li [36]. That is, $w = 0$ represents the wealth preference mechanism, i.e., the processes of investment and payoff allocation are determined by the previous income and the current investment, respectively. $w = 0.5$ represents the mixed preference mechanism, that is, the processes of investment and payoff allocation are determined by the previous income, the degree value, the current investment and the intensity of the cooperation. and $w = 1$ represents the social preference mechanism, that is, the processes of investment and payoff allocation are determined by the degree value and the intensity of the cooperation, respectively.

## 4.3 Impact of the fairness preference and return intensity on the enterprise cooperation level in cluster innovation network evolution

Fig 2 reflects the influence of the adjustment coefficient on the enterprise cooperation level in the cluster innovation network under three mechanisms and different gain coefficients. Observe in Fig 2 that under the wealth preference mechanism, with the increase in the adjustment coefficient, the level of enterprise cooperation in the network shows a trend of first decreasing, then rising, and finally stabilizing. When we have the adjustment coefficient $\alpha \leq 1$, along with the decrease of the difference degree of the previous income, degree value, current

**Table 1. Parameter settings for the evolutionary game simulation of cluster innovation networks.**

| Number of games | Network Size | Average Degree | Maximum Node Degree | Minimum Node Degree | Maximum Cooperation Strength | Minimum Cooperation Strength |
|---|---|---|---|---|---|---|
| 500 | 100 | 4 | 31 | 2 | 10 | 1 |

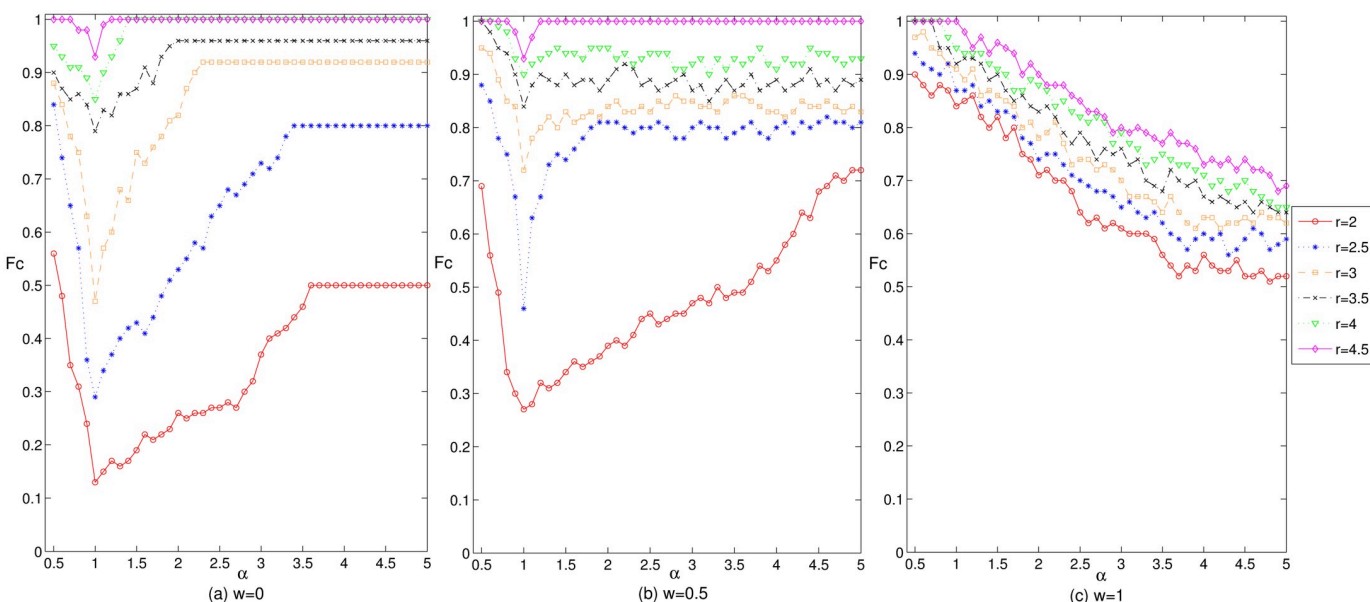

**Fig 2. The effect of the adjustment coefficient on the level of enterprise cooperation in cluster innovation network under three mechanisms and different gain coefficients.**

investment and cooperation intensity, the cooperation level of the enterprises in the network gradually decreases. Correspondingly, when we have the adjustment coefficient $\alpha > 1$, with the increase of the difference degree of the previous income, degree value, current investment and cooperation intensity, the cooperation level of the enterprises in the network gradually increases. The increase in the degree of differentiation of the previous income and current investment can promote the level of enterprise cooperation. Compared with the wealth preference mechanism, the change of enterprise cooperation level under the mixed preference mechanism is more volatile. The reason is that under the mixed preference mechanism, the linked processes of investment and payoff allocation are affected by many factors, which makes enterprises more willing to change the existing cooperative relationship. Under the effects of the reconnection mechanism, the network structure will change greatly. When the cluster innovation network structure changes greatly, the changes will make the processes of the enterprise's investment and payoff allocation more complex, which affects the choice of the enterprise cooperation strategy and leads to a greater fluctuation in the level of enterprise cooperation. Under the social preference mechanism, with the increase in the adjustment coefficient, the level of enterprise cooperation shows a gradual downward trend. This result is due to the increase in differentiation with regard to the breadth and depth of the enterprise cooperation, which renders more enterprises with less cooperation breadth and depth dissatisfied with the existing earnings. As a result, these enterprises change their partners, so the network structure changes. Whenever the network structure changes greatly, the profit margin of more enterprises is narrowed, which destroys the cooperative environment in the cluster. Thus cluster enterprises gradually withdraw from cooperation, which inhibits the improvement of the level of enterprise cooperation within the cluster [25]. In addition, the greater the differentiation is in terms of the breadth and depth of the enterprise cooperation, the more significant the interactions between the network structure and enterprise cooperative behaviors are, and the greater the inhibitory effect on the enterprise cooperation level is. In a the real cluster innovation network, the processes of enterprise investment and payoff allocation are affected by

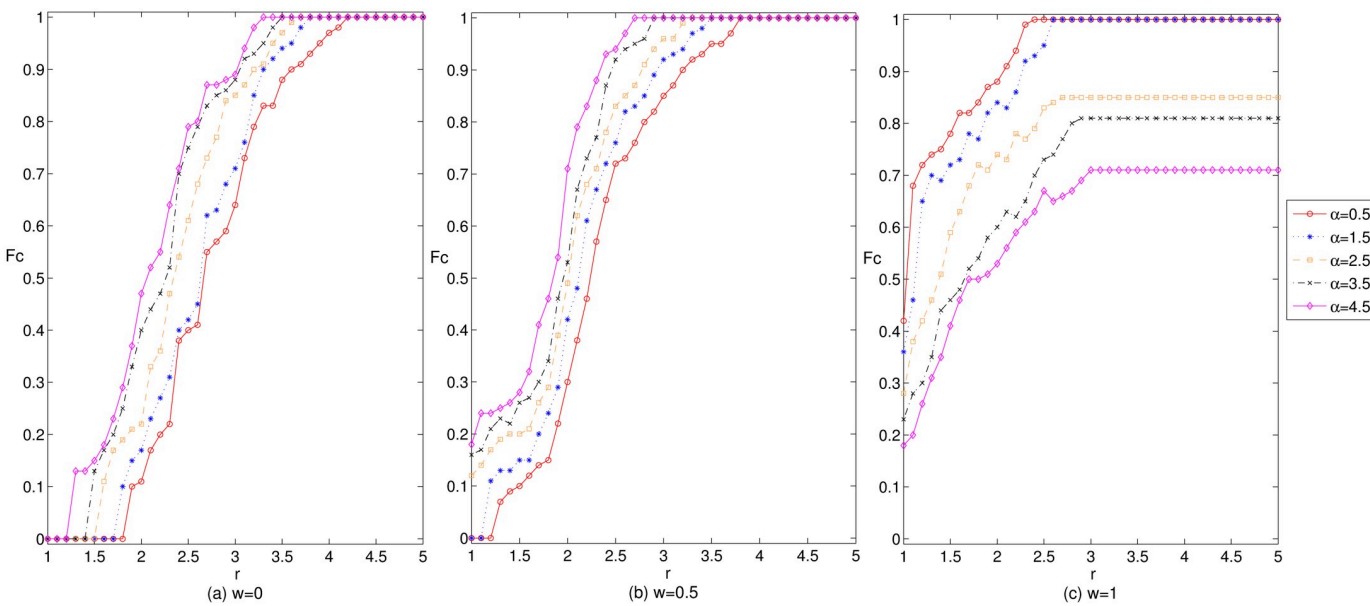

**Fig 3. The effect of the gain coefficient on the enterprise cooperation level in cluster innovation network under three mechanisms and different adjustment coefficients.**

many factors. Furthermore, excessive profit-seeking of enterprises in the processes of investment and payoff allocation is not conducive to the maintenance of cooperative relationships among the enterprises in the network, which shows why it is difficult to have a high adjustment coefficient in the game model and in the mixed preference mechanism. Therefore, the appropriate return intensity is very important to maintaining the level of enterprise cooperation in the network and to promoting the healthy evolution and development of cluster innovation networks.

Fig 3 shows the effect of the gain coefficient on the enterprise cooperation level in cluster innovation networks under three mechanisms and different adjustment coefficients. Observe in Fig 3 that under the wealth preference mechanism, with an increase in the gain coefficient, the level of enterprise cooperation in the network first shows a stable trend, then the trend rises and finally stabilizes. This observation demonstrates that the level of enterprise cooperation in the network is only promoted when the return intensity exceeds a certain threshold, and the stronger the return intensity will be, the more obvious the promotion effect on the level of enterprise cooperation is. Under the mixed preference and social preference mechanisms, the level of enterprise cooperation shows a trend of first rising and then stabilizing. However, compared with the wealth preference mechanism, there is a lack of an initial stable development stage, and when the gain coefficient is relatively low, the level of enterprise cooperation can be stabilized. This result shows that the improvement of the preferences in terms of the breadth and depth of cooperation will enhance the ability of the investment and benefit allocation in cluster enterprises. Therefore, this improved preference will replace the role of returns to some extent. In addition, when the adjustment coefficient $\alpha$ is relatively low, with an increase of the weight coefficient $w$, the level of enterprise cooperation in the network gradually increases. This result may be due to the improvement of the preferences in terms of the breadth and depth of the enterprise cooperation and the increasing position of some enterprises in the network, whose partners are more willing to choose cooperation strategies to achieve greater benefits. At the same time, these enterprises will be dissatisfied because of the

"speculative" and "free-rider" behavior of some partners and thereby change their own cooperative behavior and partners, which would cause the network structure to change greatly. When the network structure changes greatly, more enterprises will change their payoffs. Enterprises that employ non cooperative strategies will change their cooperative behavior for greater profits and, ultimately, improve the level of enterprise cooperation gradually. In addition, with an improvement of the preferences in terms of the breadth and depth of the enterprise cooperation, the interaction between cooperative behavior of innovators and network structure is stronger, and enterprise cooperation is more strongly promoted, which is more conducive to the evolution and development of the cluster innovation network. Therefore, to promote the emergence of cooperative behavior in cluster innovation networks, consider the return situation in the network in advance. When the return intensity is relatively low, the appropriate increase of the weight coefficient *w* can effectively maintain the level of enterprise cooperation at a higher level. Correspondingly, when the return intensity is relatively high, the lower weight coefficient *w* can ensure effective improvement in the enterprise cooperation.

## 4.4 Impact of the fairness preference and return intensity on the corporate cooperative benefit in cluster innovation networks

Figs 4 and 5 reflect the respective impacts of the adjustment and gain coefficients on the cooperative benefit of enterprises with different cooperative breadths in cluster innovation networks under three mechanisms. It can be seen from Figs 4 and 5 that with an increase in the weight coefficient, the profitability of enterprises with a large cooperation scale is gradually enhanced, while the cooperation income of enterprises with smaller cooperation breadth gradually decreases. The result is the phenomenon of "Care for this and lose that", furthermore income imbalance among enterprises in the network becomes increasingly apparent. At the same time, the degree of enterprises with a wide range of cooperation is increasing, and the quantity is also increasing. This pattern occurs is because with the increase in the preference in terms of the breadth and depth of enterprise cooperation, the leading enterprises exhibiting

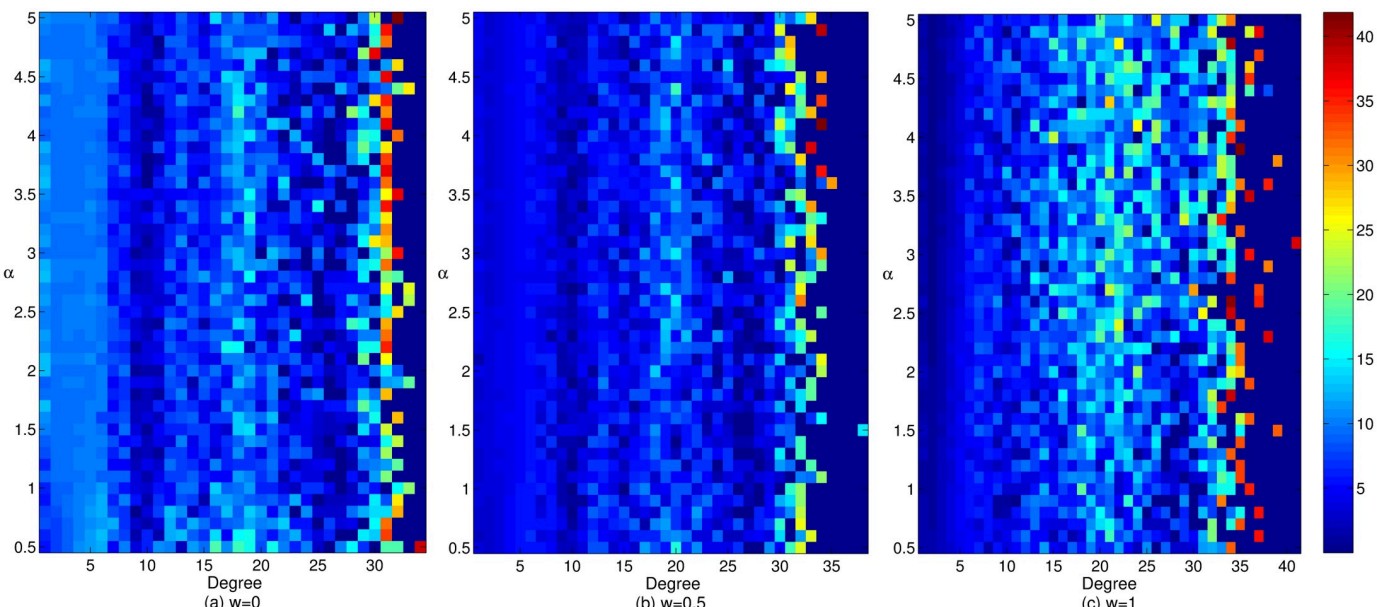

**Fig 4. The effect of the adjustment coefficient on the cooperative benefits of enterprises with different cooperative breadths in cluster innovation network under three mechanisms.**

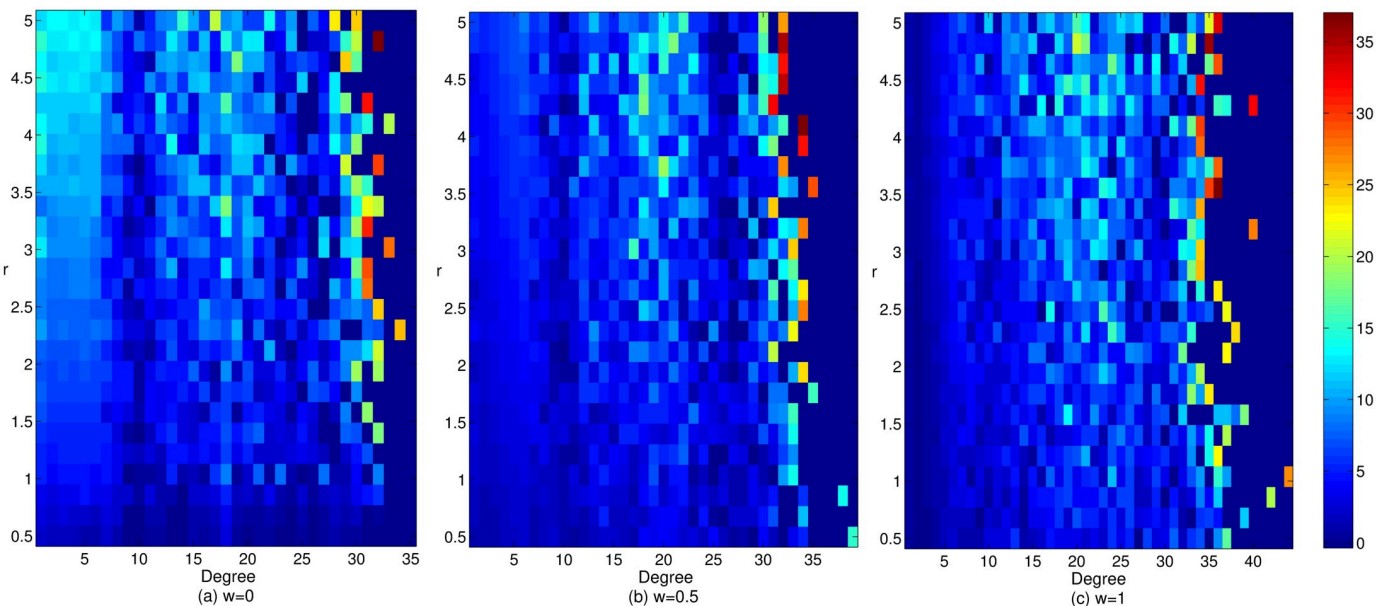

**Fig 5. The effect of the gain coefficient on the cooperative benefits of enterprises with different cooperative widths in cluster innovation networks under three mechanisms.**

broad cooperation have more substantial processes of profit distribution, which makes the cooperation income of larger enterprises increase continuously and reduce the profit margins of smaller enterprises. Therefore, the cooperation income of enterprises with smaller cooperation scope is declining continuously. In this case, the enterprises with smaller cooperation breadth will be dissatisfied because of their low income, hence, they will change their cooperative behavior. Under the effect of the reconnection mechanism, these enterprises are more inclined to R&D with the enterprises that have larger cooperation breadth to promote the increasing number of enterprises with larger cooperation scope, and their ability to benefit is also increasing gradually. In addition, with the changes in the network structure, the income of more enterprises are also changing. Increasing numbers of enterprises with smaller cooperation breadth will change their cooperative behavior because of dissatisfaction with their income. However, due to the unfavorable environment of cluster cooperation, under the interaction between cooperative behavior of innovators and the network structure, the benefits of enterprises with smaller cooperation breadth are gradually declining. Therefore, the increase in the preference in terms of the breadth and depth of the enterprise cooperation has a restraining effect on the increase in the incomes of enterprises with smaller cooperation breadth and a promoting effect on the increase in the incomes of enterprises with larger cooperation breadth, which is not conducive to the common development of all types of enterprises in cluster innovation network. The graph also reveals that with the increase in the gain coefficient $r$, the cooperative income has also increased. The enterprises with larger cooperation breadth have higher cooperative benefit. This result shows that the return intensity has a positive effect on the promotion of the corporate cooperation income, and this effect particularly significant for the promotion of enterprise cooperation income of enterprises with larger cooperation breadth. When the return intensity is continuously enhanced, this effect can effectively promote the enthusiasm of the cooperation among various types of the enterprises in the network, especially to enhance the cooperation incomes of enterprises with larger cooperation breadth to promote the benign evolution and development of cluster innovation network.

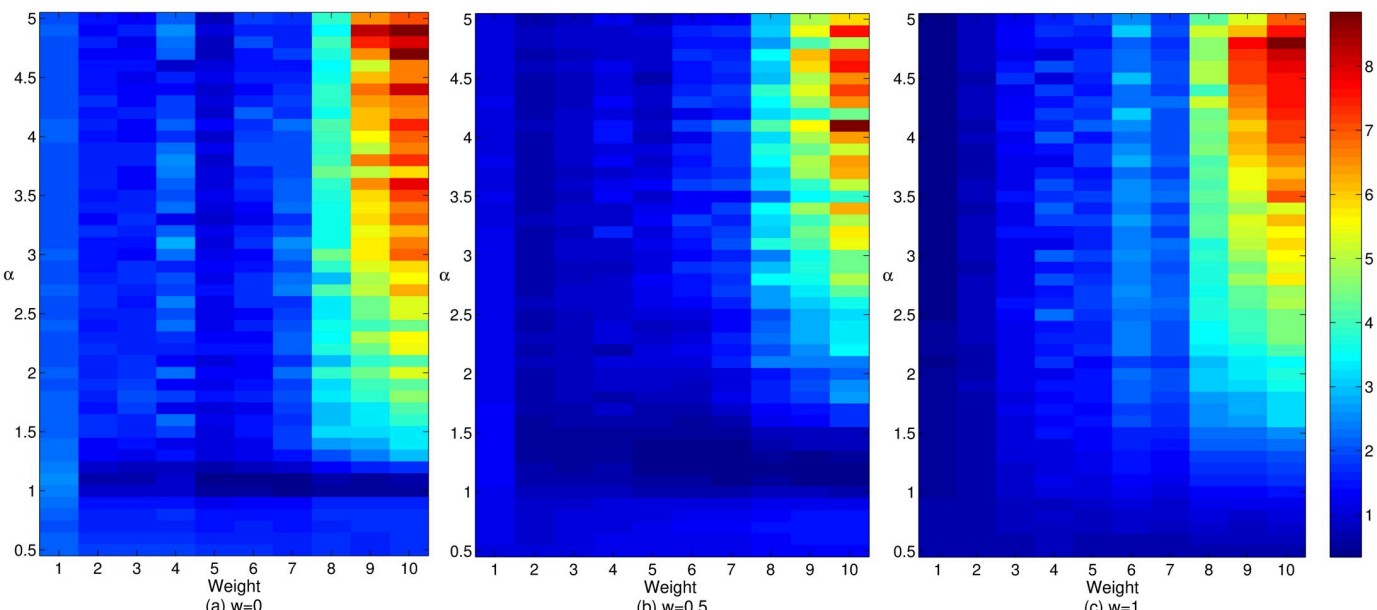

**Fig 6. The effect of the adjustment coefficient on the cooperative returns of enterprises with different cooperative depths in cluster innovation network under three mechanisms.**

Figs 6 and 7 show the respective impacts of the adjustment coefficient and gain coefficient on the cooperation income of enterprises with different cooperation depths in cluster innovation network under three mechanisms. Figs 6 and 7 illustrate that among the three mechanisms, the cooperative benefit of enterprises with smaller cooperation depth under a wealth preference mechanism is relatively higher, and the corresponding benefit of enterprises with

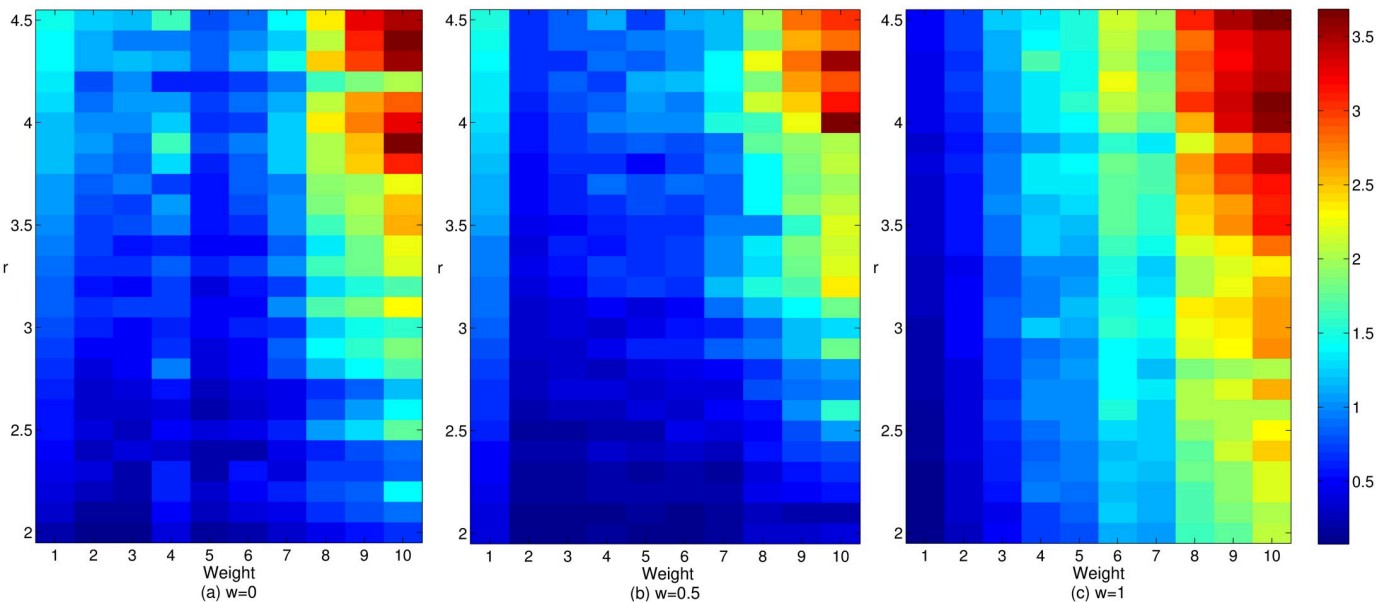

**Fig 7. The effect of the gain coefficient on the cooperative benefits of enterprises with different cooperation depths in cluster innovation network under three mechanisms.**

a larger cooperation depth is relatively lower. Under the social preference mechanism, the earnings of enterprises with smaller cooperation depth are relatively low, while those with larger cooperation depth are relatively high. This result may be due to the increase in the preference in terms of the breadth and depth of the enterprise cooperation, which enhances the enthusiasm of enterprises with larger cooperation depth for cooperation and innovation and reduces the same enthusiasm of enterprises with smaller cooperation depth. Under the effect of the reconnection mechanism, the same types of enterprises in cluster innovation network tend to cooperate, which makes the profits of enterprises with larger cooperation depth increase continuously, while those with smaller cooperation depth decrease gradually. The result is polarization, which is not conducive to the evolution and development of cluster innovation network. However, the change in the network structure will change many enterprises profits, and thereby aggravate the polarization phenomenon and increase the income gap between enterprises with larger cooperation depth and those with smaller cooperation depth. From the figure, we can also see that the cooperative benefit of both smaller and larger cooperation depth enterprises are rising in cooperation networks with the increase in the gain coefficient $r$. This result shows the increase of return can be promoted by improving the investment and profit ability of both smaller-depth and larger-depth cooperation enterprises, which will in turn prompt cooperative benefits in both smaller and larger depth cooperation enterprises. In addition, with the increase in the adjustment coefficient $\alpha$, the earnings of enterprises with larger cooperation depth are gradually increasing. This result is due to the increase in the difference degree of the previous income, the current investment, the cooperation breadth and the cooperation depth, which enlarges the gap between the rich and the poor among the enterprises in the cluster. This difference in the degree can effectively enhance the position and power of enterprises with larger cooperation depth in the process of investment and payoff allocation to give full play to their knowledge transfer and cooperation innovation, this difference can also facilitate the promotion of enterprises cooperative benefit when they have larger cooperation depth. Therefore, the enhancement of the interenterprise cooperation relationship is conducive to the promotion of the corporate cooperative benefit, which is crucial for the benign evolution and development of cluster innovation network.

## 5 Conclusions

This paper takes the cluster innovation network constructed by the weighted evolutionary BBV model as the research subject. Based on network evolutionary game theory and fairness preference theory, this study constructs a cluster innovation network evolutionary game model that includes the investment index, benefit index, cooperation breadth index and cooperation depth index. Using simulation analysis, the cooperation level and cooperation income in the evolution process of the cluster innovation network under fair preference and return intensities are analyzed, and the following conclusions are drawn.

First, increases in the weight coefficient, the gain coefficient, the difference degree of the previous income and the current investment can promote the level of enterprise cooperation. The greater the degree of the difference of the weight coefficient, the gain coefficient, the previous income, and the current investment are, the more obvious the promotion effect of the enterprise cooperation level is. Among these factors, the level of enterprise cooperation can only be promoted if the gain coefficient reaches a certain threshold. In addition, the increase in the degree of the differentiation in the terms of the breadth and depth of enterprise cooperation has a restraining effect on the enterprises cooperative level. Moreover, the greater the differentiation is, the stronger the inhibitory effect will be.

Second, the increase in the adjustment coefficient and the gain coefficient can promote increases in the cooperative income of enterprises, and the larger the adjustment coefficient and the gain coefficient are, the more obvious the promotion of the cooperative income of enterprises will be. The increase in the weight coefficient has a restraining effect on the increase in the cooperation income of enterprises with smaller cooperation breadth and deeper cooperation depth and has a promoting effect on the increase in the cooperation income of enterprises with larger cooperation breadth and deeper cooperation depth. The result is an unbalanced development mode of "Care for this and lose that" that is not conducive to the common development of cluster enterprises. In addition, the increase in the enterprise cooperation breadth and the enhancement of the interenterprise cooperation relationship can effectively promote the improvement of the enterprise cooperative benefit.

## Author Contributions

**Formal analysis:** Chuanyun Li.

**Writing – original draft:** Chuanyun Li.

**Writing – review & editing:** Xia Cao, Ming Chi.

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
