## [Decision Letter · Decision Letter 0]

12 Sep 2019

PONE-D-19-22374

Research on an Evolutionary Game Model and Simulation of a Cluster Innovation Network Based on Fairness Preference

PLOS ONE

Dear Dr. Cao,

Thank you for submitting your manuscript to PLOS ONE. After careful consideration, we feel that it has merit but does not fully meet PLOS ONE’s publication criteria as it currently stands. Therefore, we invite you to submit a revised version of the manuscript that addresses the points raised during the review process.

We would appreciate receiving your revised manuscript by Oct 27 2019 11:59PM. To enhance the reproducibility of your results, we recommend that if applicable you deposit your laboratory protocols in protocols.io, where a protocol can be assigned its own identifier (DOI) such that it can be cited independently in the future. For instructions see: http://journals.plos.org/plosone/s/submission-guidelines#loc-laboratory-protocols

We look forward to receiving your revised manuscript.

Kind regards,

Luo-Luo Jiang, Ph.D.

Academic Editor

PLOS ONE

Journal Requirements:

Reviewers' comments:

Reviewer's Responses to Questions

**Comments to the Author**

1. Is the manuscript technically sound, and do the data support the conclusions?

Reviewer #1: Partly

Reviewer #2: Partly

2. Has the statistical analysis been performed appropriately and rigorously? 

Reviewer #1: Yes

Reviewer #2: Yes

3. Have the authors made all data underlying the findings in their manuscript fully available?

Reviewer #1: Yes

Reviewer #2: Yes

4. Is the manuscript presented in an intelligible fashion and written in standard English?

Reviewer #1: No

Reviewer #2: Yes

5. Review Comments to the Author

Reviewer #1: In this manuscript, the authors concern a cluster innovation network based on based on game theory and the fair preference theory, and investigate cooperation level and cooperation income in evolution process of cluster innovation network subjected to fair preference and return intensity. Although the numerical results are interesting, I do not recommend the manuscript for publication in its present form. I can reconsider my decision if a number of issues listed below are properly addressed.

1.What is the BBV model? This manuscript investigates cluster innovation network constructed by BBV model, but there is no introduction about the model in the INTRODUCTION.

2.The first sentence in 3.2 Construction of Game Model shows that whatever the strategy Sx chooses, the investment of enterprise x is always zero. Please check it.

3.Figure.2(b) shows a nonmonotonic behavior of fc vs. α at r=2, while there is a monotonic behavior in Figure.3(b) for r=2. The results are inconsistent. Please check the numerical results.

4.The manuscript is written very poorly, strongly lacking clarity in presentation. For example:

(1)In Page2, “....and have find that in...” → “....and have found that in...”.

(2)In Page9, “...the greater the inhibitory effect on enterprise cooperation level are...” → “...the greater the inhibitory effect on enterprise cooperation level is...”.

(3)In Page10, “...improvement of preference in the terms of the breadth...” → “...improvement of preference in terms of the breadth...”.

(4)There have some inaccurate subscripts in the Eqs.(8) and (9).

(5)Some sentences are repeated, i.e., “...Sx is the enterprise x game strategy ( Sx=1 means cooperation, Sx means non-cooperation). ..”, which has been mentioned in the first paragraph in Construction of Game Model.

Please check the full manuscript. A through improvement of English is necessary.

Reviewer #2: Review report for “Research on an Evolutionary Game Model and Simulation of a Cluster Innovation Network Based on Fairness Preference” by Li et al.

This paper studies an evolutionary game model of a cluster innovation network based on a spatial public goods game and the theory of fairness preference, where the network is constructed by the weighted evolutionary BBV model. Using network-based simulations, the authors found that an increase in the weight coefficient, gain coefficient and degree of differentiation between previous income and current investment can effectively promote improvements in the level of enterprise cooperation. Moreover, the increase in regulation and gain coefficients can promote enterprise cooperation while the increase in the differentiation in the breadth and depth of enterprise cooperation will hinder enterprise cooperation.

The cooperative environment of cluster innovation networks is an important issue that deserves further investigation. I found that this paper considers an interesting problem, the structure is well organized, and the paper is well written. However, there are some issues that hinder the acceptance of this paper in its current form. In the following, I would like to give some comments and suggestions, which may help the authors improve the quality of their paper.

1, In the abstract, the authors directly go the introduction of methods and results. However, I think some introduction of general background regarding this topic should be added. Moreover, the authors are suggested to introduce their motivations after the background introduction. Furthermore, at the end of the abstract, it would be better to highlight the implication and application of their results.

2, In the introduction section, the authors present that “cluster innovation networks have the characteristics of a weighted scale-free network[8], so a simulation network based on the weighted evolution BBV model can simulate the real cluster innovation network well[9]”. I think it is not clear enough to the readers that whether cluster innovation networks have the characteristics of a weighted scale-free network. It would be better if the authors can explain more regarding this point.

3, On page 2, the authors present that “Some scholars also find that fixed static network structures such as rule networks[16], world networks[17] and scale-free networks[18] can also promote cooperation under certain conditions[19]”. I think “world networks[17]” should be “small-world networks[17]”. Moreover, in the follow sentence, “Santos[20] also found that the fairness preference of innovative subject cooperative behavior can greatly affect the level of cooperation in BA network”. It should be written as “Santos et al. [20] also found that xxx”.

4, In Figure, the sentences “Previous Income、Cooperation Breadth” and “Current Investment、Cooperation Depth” should be revised. In English, there is no “、” but “,”. The authors should replace “、” by “,” in these two sentences.

5, In section 3.2 Construction of Game Model, the authors present that “it participates in $k_x + 1$ neighborhoods centered on itself and neighbors, and its total investment is 0”. I think the total investment is 1 instead of 0. The authors should check this.

6, Before Equation (6), the sentence “as the center and directly connected enterprises) according to formula (6)”. I think Equation is more usually used than formula, and formula (6) should be replaced by “Equation (6)”. The same applies to the following sections, such as “Formula (7) shows that” and “Formula (8) shows that”. The authors are suggested to go through the paper and fix this problem.

7, Below Equation (7), the sentence “Among these variables, $\\bar{D_{x,y}}(t_n)$is the normalized value of $D_{x,y}(t_n) $, $t_n$ is the number of rounds of the public goods game (only if all nodes in the network have a round of game to end the round of game)”. The readers may wonder which kind of normalization that the authors used. I thinks it would be better to add more details.

8, For the following sentences, “the revenue of enterprise x(i)” and “denotes the degree value of enterprise y(i)”, it would be better to separate them into two sentences. It would be hard to distinguish x(i) from x and i.

9, In section 4.1 Simulation steps, the authors set an initial cooperation level as of 50%, i.e., the network cooperation density being 0.5. The authors may wonder how this parameter affects the results, and what happens if the initial cooperation level is set as 40% or 60%. Moreover, the cluster innovation network is with N=100 nodes, where I wonder how the network size affects the results. Maybe the authors want to add one or more figures to show the network size effects.

10, In Figure 2 and Figure 3, I found a lot of fluctuations in the results. The authors present that “each data point is the average of simulation results after 50 independent experiments”. I think it would be better to increase the times of independent experiments. Maybe 500 is a better choice to avoid fluctuations and make the results more reliable. Moreover, the resolution of these two figures should be remarkably enhanced, and more detailed captions should be added to explain these figures.

11, For the sentence “this paper uses Ruguo Fan [34] and Li [39] as references for the study of the weight coefficient, and implements three mechanisms to correspond to the three weight coefficients”. I think it would be better to revise “Ruguo Fan [34] and Li [39]” because it is not a typical reference style.

12, In the reference section, many papers published in Chinese are referred, such as Refs [1],[7],[8],[17], [23],[33],and [35], however, these papers are usually invisible to the international community. I wonder if these works are follow ups of some very related international studies, and some of them can be replaced by references published in English journals. I will let the authors to decide.

6. PLOS authors have the option to publish the peer review history of their article (what does this mean?). If published, this will include your full peer review and any attached files.

Reviewer #1: No

Reviewer #2: No

---

## [Author Response · Author response to Decision Letter 0]

26 Oct 2019

Reviewer #1: In this manuscript, the authors concern a cluster innovation network based on based on game theory and the fair preference theory, and investigate cooperation level and cooperation income in evolution process of cluster innovation network subjected to fair preference and return intensity. Although the numerical results are interesting, I do not recommend the manuscript for publication in its present form. I can reconsider my decision if a number of issues listed below are properly addressed.

1.What is the BBV model? This manuscript investigates cluster innovation network constructed by BBV model, but there is no introduction about the model in the INTRODUCTION.

Answer: The evolution model of weighted network was first established by Barrat, Barthelemy and Vespignani in 2004. It is usually called BBV model. It is a simple weight driven dynamic model, and generates statistical properties very similar to the real weight network, such as the law of network weight evolution with time, and the scale-free characteristics of strength distribution. Since the weighted scale-free network constructed by the weighted evolutionary BBV model was proposed, just like the BA scale-free network, it is gradually known. So in the introduction, the author did not introduce the BBV model in detail. In order to explain more clearly why BBV model is used to build cluster innovation network, the author makes the following modifications to this part of the text:

In addition, in the actual network evolution game process, the cooperation behavior of innovators is not only influenced by the network structure but also closely related to the intensity of the cooperative relationship between innovators[6], that is, the connection in the real network has the weight attribute. According to the existing research, a real network with connected weights has both the power-law distribution characteristics of the degree distribution and the power-law distribution characteristics of the strength distribution[7].

2.The first sentence in 3.2 Construction of Game Model shows that whatever the strategy Sx chooses, the investment of enterprise x is always zero. Please check it.

Answer: according to the reviewer’s opinion, the author found the writing error here, and has modified the paragraph as follows:

In the first round (tn=1 ) of the public goods game, the degree value of enterprise s is kx , so if the enterprise x chooses uncooperative strategy (Sx=0 ), the investment of enterprise x in its neighborhood is 0; if enterprise x chooses a cooperative strategy (Sx=0), it participates in Kx+1 neighborhoods centered on itself and neighbors, and its total investment is evenly distributed among Kx+1 neighborhoods.

3.Figure.2(b) shows a nonmonotonic behavior of fc vs. a at r=2, while there is a monotonic behavior in Figure.3(b) for r=2. The results are inconsistent. Please check the numerical results.

Answer: according to the reviewer’s opinion, author remade the simulation, redraws Figure 2 and figure 3, and corrects Figure 2 (b). See the manuscripts Fig2 and Fig3 for the new simulation results

4.The manuscript is written very poorly, strongly lacking clarity in presentation. For example:

(1)In Page2, “....and have find that in...” → “....and have found that in...”. 

(2)In Page9, “...the greater the inhibitory effect on enterprise cooperation level are...” → “...the greater the inhibitory effect on enterprise cooperation level is...”. 

(3)In Page10, “...improvement of preference in the terms of the breadth...” → “...improvement of preference in terms of the breadth...”. 

(4)There have some inaccurate subscripts in the Eqs.(8) and (9).

(5) Some sentences are repeated, i.e., “... is the enterprise x game strategy ( Sx=1 means cooperation, Sx=0 means non-cooperation)...”, which has been mentioned in the first paragraph in Construction of Game Model. Please check the full manuscript. A through improvement of English is necessary.

Answer: according to the reviewer’s opinion, the author revised (1)-(3) in the manuscript. For example: the previous income of the players[30] and have found that in a static BA network….

the more significant the interactions between network structure and enterprise cooperative behaviors are, and the greater the inhibitory effect on enterprise cooperation level is.

This result shows that the improvement of preference in terms of the breadth and depth of cooperation…

Comments on Revision (4), The author referred to several references and rechecked formulas (8) and (9). No editorial errors were found in the subscripts of formulas (8) and (9). Each subscript is described here. In formula (8), The author referred to several references and rechecked formulas (8) and (9). No editorial errors were found in the subscripts of formulas (8) and (9). Each subscript is described here. In formula (8), the subscript Mxy represents the income of x in neighborhood y , the subscript Ixy represents the input of x in neighborhood y, and the subscript y=0 in formula (9) represents the neighborhood centered on x (neighborhood y refers to the neighborhood composed of neighbor enterprise y and its directly connected enterprises). If you have any questions from the reviewer, please point out and I will revise it.

Comments on Revision (5), after careful comparison, the author has deleted similar sentences in the text and asked AJE to polish the full text, so as to improve the English level of the manuscript. Red font in English Revision

Reviewer #2: Review report for “Research on an Evolutionary Game Model and Simulation of a Cluster Innovation Network Based on Fairness Preference” by Li et al.

The cooperative environment of cluster innovation networks is an important issue that deserves further investigation. I found that this paper considers an interesting problem, the structure is well organized, and the paper is well written. However, there are some issues that hinder the acceptance of this paper in its current form. In the following, I would like to give some comments and suggestions, which may help the authors improve the quality of their paper.

1, In the abstract, the authors directly go the introduction of methods and results. However, I think some introduction of general background regarding this topic should be added. Moreover, the authors are suggested to introduce their motivations after the background introduction. Furthermore, at the end of the abstract, it would be better to highlight the implication and application of their results.

Answer: according to the reviewer’s opinion, the author added the background and purpose of the writing topic in the abstract, and added the significance of the paper research in the end. The added content is as follows:

At the beginning of the abstract: The cluster innovation network is an important part of regional economic development. In addition, the fairness preference of internal innovators in the processes of investment and benefit distribution are particularly important for curbing "free riding" and other speculative behaviors and for creating a good cooperation environment. Therefore, taking……

At the end of the abstract: Finally, we hope to improve the overall cooperation level and cooperation income of the network by deeply understanding the fair preferences of innovators in the processes of investment and benefit distribution, which is helpful for promoting the evolution and development of cluster innovation networks.

2, In the introduction section, the authors present that “cluster innovation networks have the characteristics of a weighted scale-free network[8], so a simulation network based on the weighted evolution BBV model can simulate the real cluster innovation network well[9]”. I think it is not clear enough to the readers that whether cluster innovation networks have the characteristics of a weighted scale-free network. It would be better if the authors can explain more regarding this point.

Answer: according to the reviewer’s opinion, in order to explain more clearly why BBV model is used to build cluster innovation network, the author makes the following modifications to this part of the text:

In addition, in the actual network evolution game process, the cooperation behavior of innovators is not only influenced by the network structure but also closely related to the intensity of the cooperative relationship between innovators[6], that is, the connection in the real network has the weight attribute. According to the existing research, a real network with connected weights has both the power-law distribution characteristics of the degree distribution and the power-law distribution characteristics of the strength distribution[7].

3, On page 2, the authors present that “Some scholars also find that fixed static network structures such as rule networks[16], world networks[17] and scale-free networks[18] can also promote cooperation under certain conditions[19]”. I think “world networks[17]” should be “small-world networks[17]”. Moreover, in the follow sentence, “Santos[20] also found that the fairness preference of innovative subject cooperative behavior can greatly affect the level of cooperation in BA network”. It should be written as “Santos et al. [20] also found that xxx”

Answer: according to the reviewer’s opinion, The author found that he had made some mistakes in the translation process, and now he has made some modifications to these two problems. Amend to read: Some scholars also find that fixed static network structures such as rule networks[16], small-world networks[17] and scale-free networks[18] can also promote cooperation under certain conditions[19]. Santos et al[20] also found that the fairness preference of innovative subject cooperative behavior can greatly affect the level of cooperation in BA network.

4, In Figure, the sentences “Previous Income、Cooperation Breadth” and “Current Investment、Cooperation Depth” should be revised. In English, there is no “、” but “,”. The authors should replace “、” by “,” in these two sentences.

Answer: according to the reviewer’s opinion, the author changed "、" in Fig1 to "," and modified some words in Figure 1. Figure 1 in the manuscript shows the modification result of the figure.

5, In section 3.2 Construction of Game Model, the authors present that “it participates in Kx+1 neighborhoods centered on itself and neighbors, and its total investment is 0”. I think the total investment is 1 instead of 0. The authors should check this.

Answer: after checking, the author found that there are some problems in this sentence, and has modified the paragraph as follows:

In the first round ( tx=1) of the public goods game, the degree value of enterprise x is Kx , so if the enterprise x chooses uncooperative strategy ( Sx=0), the investment of enterprise x in its neighborhood is 0; if enterprise chooses a cooperative strategy (Sx=1 ), it participates in Kx+1 neighborhoods centered on itself and neighbors, and its total investment is evenly distributed among Kx+1 neighborhoods.

6, Before Equation (6), the sentence “as the center and directly connected enterprises) according to formula (6)”. I think Equation is more usually used than formula, and formula (6) should be replaced by “Equation (6)”. The same applies to the following sections, such as “Formula (7) shows that” and “Formula (8) shows that”. The authors are suggested to go through the paper and fix this problem.

Answer: according to the reviewer’s opinion, the author has modified the “formula” in the full text to Eqs. (Equation) Since formula 6-11 has a large span in the text, it is not listed here. The modified part has been marked with red font.

7, Below Equation (7), the sentence “Among these variables, Dxy(tn) is the normalized value of Dxy(tn) , tn is the number of rounds of the public goods game (only if all nodes in the network have a round of game to end the round of game)”. The readers may wonder which kind of normalization that the authors used. I thinks it would be better to add more details.

Answer: according to the reviewer’s opinion, The author considers that the total investment of each enterprise is 1. At the same time, in order to ensure the linear change of investment, the author uses min max normalization method to standardize. The content of this paper is modified as follows:

Among these variables, considering that the total investment of each enterprise is 1, Dxy(tn) is normalized according to the min-max normalization method. Dxy(tn) is the normalized value of Dxy(tn) , tn is the number of rounds of the public goods game (only if all nodes in the network have a round of game to end the round of game).

8, For the following sentences, “the revenue of enterprise x(i)” and “denotes the degree value of enterprise y(i)”, it would be better to separate them into two sentences. It would be hard to distinguish x(i) from x and i.

Answer: according to the reviewer’s opinion, according to the expert opinion, the author divides and adjusts the content of the sentence as follows:

…. and Mxy(tn-1) is the revenue of enterprise i from neighborhood y after the first round tn-1 of the public goods game. and Mxy(tn-1) is the revenue of enterprise x from neighborhood y after the first round t-1 of the public goods game.

9, In section 4.1 Simulation steps, the authors set an initial cooperation level as of 50%, i.e., the network cooperation density being 0.5. The authors may wonder how this parameter affects the results, and what happens if the initial cooperation level is set as 40% or 60%. Moreover, the cluster innovation network is with N=100 nodes, where I wonder how the network size affects the results. Maybe the authors want to add one or more figures to show the network size effects.

Answer: according to the reviewer’s opinion, considering the below two reasons, the author decided not to increase the network cooperation density and network scale in this paper.

（1）before the experts put forward this opinion, the author has studied the influence of initial cooperation density and network scale on cooperation behavior, but found that the research conclusions are not innovative in this paper, so they are deleted and not reflected in the text. A. the emergence of cooperative behavior in the cluster innovation network is closely related to the game environment around the innovation subject. When adjusting the network cooperation density (0.4 or 0.6), it does not have a substantial impact on the change of cooperative behavior, only prolongs the network evolution time. B. before that, I have studied the network size of 50, 100, 200 and 500 under various influencing factors. It is found that the difference of the influence of network scale on cooperation behavior is only reflected in the number of iterations (evolution time). C. due to the figures 2.3 and 2.4 doesn’t reflect the trend of cooperation level over time. Therefore, the influence of network cooperation density and network scale on cooperation behavior can’t be reflected in this paper, so these two factors are put here, which makes the innovation slightly inadequate. Considering the actual simulation time and the dynamic evolution process of the network, the author finally selected the network scale of 100 for simulation research. However, according to the review experts, the research on the relationship between network scale and cooperation behavior over time has been reflected in another article "the game simulation research on knowledge transfer evolution of cluster innovation network under different network scales".

(2) this paper focuses on the impact of fair preferences of investment and payoff allocation on the cooperative behavior in the evolution process of cluster innovation network. In order to avoid too scattered research, the network cooperation density and network scale are not included in this study.

10, In Figure 2 and Figure 3, I found a lot of fluctuations in the results. The authors present that “each data point is the average of simulation results after 50 independent experiments”. I think it would be better to increase the times of independent experiments. Maybe 500 is a better choice to avoid fluctuations and make the results more reliable. Moreover, the resolution of these two figures should be remarkably enhanced, and more detailed captions should be added to explain these figures.

Answer: according to the reviewer’s opinion, and due to the limited simulation time, the author increased the number of independent experiments to 200 times, and took the average value of simulation results after 200 times. Therefore, the author re simulated, The new simulation results are shown in Fig2 and Fig3 in the manuscript.

11, For the sentence “this paper uses Ruguo Fan [34] and Li [39] as references for the study of the weight coefficient, and implements three mechanisms to correspond to the three weight coefficients”. I think it would be better to revise “Ruguo Fan [34] and Li [39]” because it is not a typical reference style.

Answer: according to the reviewer’s opinion, the author has deleted this part as required. 

12, In the reference section, many papers published in Chinese are referred, such as Refs [1],[7],[8],[17], [23],[33],and [35], however, these papers are usually invisible to the international community. I wonder if these works are follow ups of some very related international studies, and some of them can be replaced by references published in English journals. I will let the authors to decide.

Answer: The author deleted the literature published in Chinese [1], [23], [33]. Chinese literature [5], [7], [8], [14], [17] and [35] are replaced by similar references published in some English journals, in order to meet the requirements of English journals.

---

## [Decision Letter · Decision Letter 1]

11 Nov 2019

PONE-D-19-22374R1

Research on an Evolutionary Game Model and Simulation of a Cluster Innovation Network Based on Fairness Preference

PLOS ONE

Dear Dr. Cao

Thank you for submitting your manuscript to PLOS ONE. After careful consideration, we feel that it has merit but does not fully meet PLOS ONE’s publication criteria as it currently stands. Therefore, we invite you to submit a revised version of the manuscript that addresses the points raised during the review process.

We would appreciate receiving your revised manuscript by Dec 26 2019 11:59PM. To enhance the reproducibility of your results, we recommend that if applicable you deposit your laboratory protocols in protocols.io, where a protocol can be assigned its own identifier (DOI) such that it can be cited independently in the future. For instructions see: http://journals.plos.org/plosone/s/submission-guidelines#loc-laboratory-protocols

We look forward to receiving your revised manuscript.

Kind regards,

Luo-Luo Jiang, Ph.D.

Academic Editor

PLOS ONE

Reviewers' comments:

Reviewer's Responses to Questions

**Comments to the Author**

1. If the authors have adequately addressed your comments raised in a previous round of review and you feel that this manuscript is now acceptable for publication, you may indicate that here to bypass the “Comments to the Author” section, enter your conflict of interest statement in the “Confidential to Editor” section, and submit your "Accept" recommendation.

Reviewer #1: All comments have been addressed

Reviewer #2: (No Response)

2. Is the manuscript technically sound, and do the data support the conclusions?

Reviewer #1: Yes

Reviewer #2: Yes

3. Has the statistical analysis been performed appropriately and rigorously? 

Reviewer #1: Yes

Reviewer #2: Yes

4. Have the authors made all data underlying the findings in their manuscript fully available?

Reviewer #1: Yes

Reviewer #2: Yes

5. Is the manuscript presented in an intelligible fashion and written in standard English?

Reviewer #1: Yes

Reviewer #2: Yes

6. Review Comments to the Author

Reviewer #1: The revised manuscript reads much better, is significantly improved and looks suitable for publication in Plos One.

Reviewer #2: Review report for “Research on an Evolutionary Game Model and Simulation of a Cluster Innovation Network Based on Fairness Preference”.

I would thank the authors for considering my previous comments and suggestions in revising their manuscript. I think the new version has been well improved. Before the consideration for publication, however, the authors are suggested to address the following two issues.

1. For all figures, the labels of axis (as well as titles) are currently so small that it is very hard for the readers to pick up the information. The authors are suggested to remarkably increase the font size in all figures.

2. Regarding my previous comment (9) “In section 4.1 Simulation steps, the authors set an initial cooperation level as of 50%, i.e., the network cooperation density being 0.5. The authors may wonder how this parameter affects the results, and what happens if the initial cooperation level is set as 40% or 60%. Moreover, the cluster innovation network is with N=100 nodes, where I wonder how the network size affects the results. Maybe the authors want to add one or more figures to show the network size effects”, I don’t think the authors tried their best to answer my question.

Firstly, in their reply, they presented that “When adjusting the network cooperation density (0.4 or 0.6), it does not have a substantial impact on the change of cooperative behavior, only prolongs the network evolution time”. However, I failed to find any figure in their reply that can support their claim.

Secondly, the authors presented that “due to the figures 2.3 and 2.4 doesn’t reflect the trend of cooperation level over time. Therefore, the influence of network cooperation density and network scale on cooperation behavior can’t be reflected in this paper”. However, I could not find “figures 2.3 and 2.4”, and I am curious about the effects of the network scale and cooperation density other than the trend of cooperation level over time.

Thirdly, the authors presented that “the relationship between network scale and cooperation behavior over time has been reflected in another article "the game simulation research on knowledge transfer evolution of cluster innovation network under different network scales".”. However, I failed to find this referred article in web of science or using Google Scholar. I wonder if there is really a paper, and if this paper should be cited in the main text.

7. PLOS authors have the option to publish the peer review history of their article (what does this mean?). If published, this will include your full peer review and any attached files.

Reviewer #1: No

Reviewer #2: No

---

## [Author Response · Author response to Decision Letter 1]

22 Nov 2019

Since the image cannot be pasted in the reply, I hope the editor can help me to provide the reviewer 2 with the word “response to reviewer”，and the figure in it can help to explain my answer.

Reviewer #2: Review report for “Research on an Evolutionary Game Model and Simulation of a Cluster Innovation Network Based on Fairness Preference”.

I would thank the authors for considering my previous comments and suggestions in revising their manuscript. I think the new version has been well improved. Before the consideration for publication, however, the authors are suggested to address the following two issues.

1. For all figures, the labels of axis (as well as titles) are currently so small that it is very hard for the readers to pick up the information. The authors are suggested to remarkably increase the font size in all figures. 

Answer: according to the reviewer’s opinion, the author enlarges the label of axis (as well as the title) font of all Figure in the manuscript to ensure that readers can easily obtain information. Due to there are many Figure in manuscript, so I do not be displayed here. See the revised manuscript for details.

2. Regarding my previous comment (9) “In section 4.1 Simulation steps, the authors set an initial cooperation level as of 50%, i.e., the network cooperation density being 0.5. The authors may wonder how this parameter affects the results, and what happens if the initial cooperation level is set as 40% or 60%. Moreover, the cluster innovation network is with N=100 nodes, where I wonder how the network size affects the results. Maybe the authors want to add one or more figures to show the network size effects”, I don’t think the authors tried their best to answer my question.

Firstly, in their reply, they presented that “When adjusting the network cooperation density (0.4 or 0.6), it does not have a substantial impact on the change of cooperative behavior, only prolongs the network evolution time”. However, I failed to find any figure in their reply that can support their claim.

Answer: according to the reviewer’s opinion, the author put research on the influence of different network cooperation density on the evolution of cluster innovation network cooperation here to support the author's explanation. To ensure the scientific nature of the research, the author sets the weight coefficient w=0.5 (hybrid preference mechanism), the adjustment coefficient ( alpha=1，alpha=2 and alpha=3 ), three kinds of gain coefficient ( r=1,r=2 and r=3 ), and three kinds of network cooperation density (p=0.4, p=0.5 and p=0.6 ) to carry out simulation analysis, to illustrate the influence of the network cooperation density on the evolution of cluster innovation network under different adjustment coefficients and gain capabilities. The simulation times are set to 100. After the system fully evolves to the stable stage, the average value of the last 30 cooperation densities is taken as P .

Figure 1 show the effect of the network cooperation density on the enterprise cooperation level in cluster innovation network under different adjustment coefficients. It can be seen from Figure 1 that under the same adjustment coefficient, with the increase of network cooperation density, the time from network evolution to stability is gradually longer, and the network evolution speed is gradually reduced. In addition, in the three adjustment coefficients, the simulation results are similar. Figure 2 show the effects of the network cooperation density on the enterprise cooperation level in cluster innovation network under different gain coefficients. the simulation results in Figure 2 are similar to those in Figure 1. Therefore, we have the same result.

Because of the similar simulation results under different parameters, the author thinks that it is not enough to study the influence of network cooperation density on the evolution of cluster innovation network cooperation. Therefore, the author thinks over and over again, and does not put the research results in the manuscript, and hopes that the reviewers will understand.

(a) alpha=1 (b) alpha=2 (c) alpha=3

Fig. 1 The effect of the network cooperation density on the enterprise cooperation level in cluster innovation network under different adjustment coefficients

(a) r=2 (b) r=3 (c) r=4

Fig. 2 The effects of the network cooperation density on the enterprise cooperation level in cluster innovation network under different gain coefficients

Secondly, the authors presented that “due to the figures 2.3 and 2.4 doesn’t reflect the trend of cooperation level over time. Therefore, the influence of network cooperation density and network scale on cooperation behavior can’t be reflected in this paper”. However, I could not find “figures 2.3 and 2.4”, and I am curious about the effects of the network scale and cooperation density other than the trend of cooperation level over time.

Answer: Due to my writing mistakes, let you have misunderstanding figure 2.3 or 2.4, I apologize to you. What I want to explain is that in Figure 2, Figure 3, … and in Figure 7, the change trend of cooperation level over time is not reflected, and the influence of network cooperation density and network scale on cooperation behavior is not reflected in the manuscript. Therefore, it will be awkward to put these two factors here alone, and it will appear that innovation is a little inadequate. 

Thirdly, the authors presented that “the relationship between network scale and cooperation behavior over time has been reflected in another article "the game simulation research on knowledge transfer evolution of cluster innovation network under different network scales".”. However, I failed to find this referred article in web of science or using Google Scholar. I wonder if there is really a paper, and if this paper should be cited in the main text.

Answer: first of all, here, the author apologizes to the reviewer. In the last reply to your opinion, I did not inform the reviewer of the relevant information of that manuscript. Here, Here, the author explains the manuscript to you. 

The manuscript “Evolutionary Game Simulation of Knowledge Transfer in Industry-University-Research Cooperative Innovation Network under Different Network Scales” is currently under reviewed in 《scientific reports》. Due to it has not been uploaded, so it cannot be found in web of science or using Google Scholar. The manuscript was submitted at the end of August, and the overhaul comments were received on October 4st. On November 21st, the author submit the revised manuscript to 《scientific reports》. Among them, according to the reviewer’s opinion, in order to highlight the difference between enterprises and research institutions in knowledge transfer, the manuscript title has been slightly adjusted. The title “Evolutionary Game Simulation of Knowledge Transfer in Cluster Innovation Network under Different Network Scales” is modified to “Evolutionary Game Simulation of Knowledge Transfer in Industry-University-Research Cooperative Innovation Network under Different Network Scales”. Figure 3 is the proof of manuscript review in 《scientific reports》 after overhaul. Since the manuscript was revised before and after, the research on the relationship between the network scale and cooperative behavior has not changed much. Therefore, the conclusions of this study can be used for reference by reviewers. Considering that the manuscript is still in the process of review, the author will only present the relevant research contents about network scale and cooperation behavior in the manuscript “Evolutionary Game Simulation of Knowledge Transfer in Industry-University-Research Cooperative Innovation Network under Different Network Scales”, please understand. In the following text, red font is the conclusion of the study. There is a discussion about the influence of network structure on the evolution speed of knowledge transfer in cooperative behavior in the text.

Fig.3 Proof of manuscript review in scientific reports

Evolutionary Game Simulation of Knowledge Transfer in Industry-University-Research Cooperative Innovation Network under Different Network Scales

Abstract: This paper takes the industry-university-research cooperation innovation network constructed by the weighted evolutionary BBV model as the research object, which is based on bipartite graph and evolutionary game theory, and constructing the game model of knowledge transfer in the industry-university-research cooperation innovation network, by using the simulation analysis method and analyzing the evolution law of knowledge transfer in the industry-university-research cooperation innovation network under different network scales, three scenarios, the knowledge transfer coefficient and the knowledge reorganization coefficient. The results show that the increase of network size reduces the speed of knowledge transfer in the network. and the greater the average cooperation intensity of the nodes, the higher the evolution depth of knowledge transfer. Compared with university-research institutes, the evolution depth of knowledge transfer in enterprises is higher, and with the increase of network scale, the gap between the evolution depth of knowledge transfer between them is gradually increasing. Only when reward, punishment and synergistic innovation benefits are higher than the cost of knowledge transfer that can promote the benign evolution of industry-university-research cooperation innovation networks. Only when the knowledge transfer coefficient and the knowledge reorganization coefficient exceed a certain threshold will knowledge transfer behavior emerge in the network. With the increase of the knowledge transfer coefficient and the knowledge reorganization coefficient, the knowledge transfer evolutionary depth of the average cooperation intensity of all kinds of nodes is gradually deepening.

Key words: BBV model; Industry-University-Research Cooperative Innovation network; knowledge transfer; game model; simulation

4.3 Simulation Results Analysis

(1) The influence of the three scenarios on the knowledge transfer evolution of the industry-university-research cooperation innovation network under different network scales.

It can be found from Fig. 4 that the evolution simulation of knowledge transfer in the industry-university-research cooperation innovation network has similar evolution results under different network scales. In the same situation, along with the increase of network scale, the time of knowledge transfer evolving to a stable stage in the industry-university-research cooperation innovation network is gradually longer, and the evolution speed of knowledge transfer is gradually slower. This may be because the node degree, average weighted degree and shortest path in small-scale networks are relatively small, and the efficiency of information transmission is high. In large-scale networks, the more cooperative relationships there are among the nodes, the more uneven distribution and heterogeneity of the nodes, the more complex the process of revenue comparison and strategy learning among the nodes in the game process, the lower the efficiency of information transmission and the slower the speed of knowledge transfer 44.

(a) n=100 (b) n=200 (c) n=500

Figure 4 Evolution simulation results of knowledge transfer in the industry-university-research cooperation innovation network under different network scales and three scenarios

(2) The influence of the knowledge transfer coefficient on the knowledge transfer evolution of the industry-university-research cooperation innovation network under different network scales.

On the basis of the situation in which the evolutionary depth of knowledge transfer in the industry-university-research cooperation innovation network is 0, the knowledge transfer coefficients of advantage university-research institutes, advantage enterprise, general university-research institutes, and general enterprise obey the uniform distribution of (0.05,0.15) ,(0.10,0.20) ,(0.15,0.25) and (0.20,0.30) in turn. The values of i are 0.0, 0.2, 0.4 and 0.6, respectively, to adjust the knowledge transfer coefficient alpha of all kinds of innovators, expressed by P1, P2, P3 and P4, respectively. The simulation results of the knowledge transfer evolution in the industry-university-research cooperation innovation network under different network scales and knowledge transfer coefficients are obtained, as shown in Fig. 5. Among them, Fig. 5 (a, b and c) respectively shows the simulation results of knowledge transfer evolution under three network scales, P1, P2, P3 and P4 represent the simulation evolution curves under four knowledge transfer coefficients. 

From Fig. 5, it can be found that when the final results of network evolution converge to 0 under the same network scale, with the increase of knowledge transfer coefficient, the time for the evolution of the industry-university-research cooperation innovation network to be stable gradually becomes longer. This is because the increase of the knowledge transfer coefficient enhances the knowledge transfer ability, the knowledge transfer willingness and the knowledge absorption ability of the innovators, and the innovators’ knowledge transfer behavior preference gradually strengthens, so that the innovators’ cooperative behavior preference changes from a "non transfer" dominant direction to a "transfer" dominant direction. Under the same knowledge transfer coefficient, along with the increase of the network scale, the time for the evolution of the industry-university-research cooperation innovation network to stability gradually becomes longer. This is due to the increase of the network scale, the average path between innovators increases gradually, and the efficiency of information transmission decreases gradually. In the process of network games, innovators are faced with more complex revenue comparison and strategy selection, which leads to the slow evolution of knowledge transfer and the time from the evolution of the industry-university-research cooperation innovation network to stabilization gradually becomes longer. In addition, it can be found that only when i is 0.6, that is, when the knowledge transfer coefficient is the maximum, the evolution result of knowledge transfer in the industry-university-research cooperation innovation network converges to 1. It shows that only when the knowledge transfer coefficient is higher than a certain threshold will the knowledge transfer behavior of innovators emerge in the network48.

(a) n=100 (b) n=200 (c) n=500

Figure 5 Evolution simulation results of knowledge transfer in the industry-university-research cooperation innovation network under different network scales and knowledge transfer coefficients

(3) The influence of the knowledge reorganization coefficient on the evolution of knowledge transfer in the industry-university-research cooperation innovation network under different network scales.

On the basis of the situation in which the evolutionary depth of knowledge transfer in the industry-university-research cooperation innovation network is 0, the knowledge reorganization coefficients of advantage university-research institutes, advantage enterprise, general university-research institutes, general enterprise obey the uniform distribution of (0.02,0.03),(0.015,0.025) , (0.01,0.02) and (0.005,0.015) in turn. The values of i are 0.01, 0.02, 0.03, and 0.04, respectively, to adjust the knowledge reorganization coefficients beta of all kinds of innovators, expressed by P1, P2, P3 and P4, respectively. The simulation results of the knowledge transfer evolution in the industry-university-research cooperation innovation network under different network scales and knowledge reorganization coefficients are obtained, as shown in Fig. 6. Among them, Fig. 6 (a, b and c) respectively shows the evolution results of knowledge transfer under three network scales. P1, P2, P3 and P4 represent the simulation evolution curves under four knowledge reorganization coefficients.

From Fig. 6, it can be found that when the final results of network evolution converge to 0 under the same network scale, with the increase of the knowledge reorganization coefficient, the time of knowledge transfer evolving to stability is gradually longer. However, when the final results of network evolution converge to 1, with the increase of the knowledge reorganization coefficient, the time of knowledge transfer evolving to stability is gradually shortened. This may be due to the increase of the knowledge reorganization coefficient, which makes the ability of innovators to understand, comprehend and apply knowledge gradually enhanced. The new knowledge acquired through digestion, absorption and reinnovation increases gradually, so that the knowledge transfer behavior preference of the innovator is gradually enhanced and then promotes the steady development of knowledge transfer in the industry-university-research cooperation innovation network. With the same knowledge reorganization coefficient, along with the increase of network scale, the time of knowledge transfer evolving to stability in the industry-university-research cooperation innovation network gradually becomes longer. This is because with the increase of the network scale, the level of knowledge application ability of innovators is not uniform, and the average path length between innovators is gradually increasing, which affects the transmission efficient knowledge information in the network, thus reducing the evolution speed of knowledge transfer and delaying the evolution time of knowledge transfer. In addition, it can be found from the graph that the final result of knowledge transfer evolution of the industry-university-research cooperation innovation network converges to 1 only when j is 0.03. This also shows that only when the knowledge reorganization coefficient is higher than a certain threshold, the innovator has a certain ability of understanding, comprehending and applying the knowledge, so that the knowledge transfer behavior will emerge in the network.

(a) n=100 (b) n=200 (c) n=500

Figure 6 Evolution simulation results of knowledge transfer in the industry-university-research cooperation innovation network under different network scales and knowledge reorganization coefficients

---

## [Editor Report · Decision Letter 2]

6 Dec 2019

Research on an Evolutionary Game Model and Simulation of a Cluster Innovation Network Based on Fairness Preference

PONE-D-19-22374R2

Dear Dr. Cao,

We are pleased to inform you that your manuscript has been judged scientifically suitable for publication and will be formally accepted for publication once it complies with all outstanding technical requirements.

With kind regards,

Luo-Luo Jiang, Ph.D.

Academic Editor

PLOS ONE
---

## [Editor Report · Acceptance letter]

27 Dec 2019

PONE-D-19-22374R2 

Research on an Evolutionary Game Model and Simulation of a Cluster Innovation Network Based on Fairness Preference 

Dear Dr. Cao:

I am pleased to inform you that your manuscript has been deemed suitable for publication in PLOS ONE. Congratulations! Your manuscript is now with our production department. 

With kind regards,

on behalf of

Dr. Luo-Luo Jiang 

Academic Editor

PLOS ONE